# Metformin Improves the Hepatic Steatosis Index in Non-Obese Patients with Polycystic Ovary Syndrome

**DOI:** 10.3390/jcm11154294

**Published:** 2022-07-24

**Authors:** Annika Riemann, Martina Blaschke, Annukka Jauho-Ghadimi, Heide Siggelkow, Katja Susanne Claudia Gollisch

**Affiliations:** 1Department of Gastroenterology, Gastrointestinal Oncology and Endocrinology, University Medical Center Göttingen, 37075 Göttingen, Germany; annika.riemann@stud.uni-goettingen.de (A.R.); martina.blaschke@medizin.uni-goettingen.de (M.B.); heide.siggelkow@amedes-group.com (H.S.); 2MVZ Endokrinologikum Göttingen, 37075 Göttingen, Germany; mari.jauho-ghadimi@amedes-group.com

**Keywords:** NAFLD, polycystic ovarian syndrome, PCOS, metformin, hepatic steatosis index

## Abstract

Non-alcoholic fatty liver disease (NAFLD) is a common yet little recognized health problem in women with polycystic ovary syndrome (PCOS). In a retrospective setting, we investigated the effects of metformin treatment on the hepatic steatosis index (HSI) as a readily available biomarker panel for NAFLD. HSI values of >36 are considered to be highly suggestive for NAFLD. In our cohort, HSI values indicating NAFLD were found in 60/81 (74.1%) women at baseline. The mean HSI improved significantly after the metformin treatment from 43.2 ± 1.0 to 41.0 ± 1.1. Subgroup analyses of non-obese (body mass index (BMI) < 30 kg/m^2^), obese (BMI 30–35 kg/m^2^) and very obese (BMI > 35 kg/m^2^) women yielded mean baseline HSI values of 35.5 ± 4.5, 41.2 ± 2.7 and 51.2 ± 4.7, respectively. A significant improvement in the HSI of 1.5 ± 2.1 was observed after metformin treatment in non-obese women but not in the obese subgroups. The data suggest a new aspect of metformin treatment in non-obese PCOS patients, namely, a possible improvement in NAFLD. This study highlighted hepatic steatosis as a common comorbidity in PCOS patients that can severely affect their long-term health, and therefore, deserves more attention in the management of PCOS patients.

## 1. Introduction

Non-alcoholic fatty liver disease (NAFLD) is one major metabolic complication in young women with polycystic ovary syndrome (PCOS). In fact, patients with PCOS are nearly four times more likely to develop NAFLD than healthy women [1]. In total, NAFLD could be found in up to 84% of PCOS patients [2]. NAFLD is defined as hepatic steatosis that is not caused by increased alcohol consumption, any other chronic liver disease (e.g., hepatitis type B or C) or medication [3]. NAFLD, which includes both non-alcoholic fatty liver (NAFL) and non-alcoholic steatohepatitis (NASH), is the most common liver disease in Western countries and affects approximately 25% of adults worldwide [4]. Since NAFLD is not only a relevant liver disease but also a major cardiovascular risk factor that may be independent of other cardiovascular risk factors [5], NAFLD should be perceived as a major health problem for women with PCOS. 

Polycystic ovary syndrome (PCOS), also known as Stein–Leventhal syndrome, is the most common metabolic disorder and a frequent cause of anovulatory infertility in young women [6]. Depending on the diagnostic tool, between 5 and 19% of all women of productive age suffer from PCOS [7]. There seem to be loco-regional differences, with higher prevalences in India, Australia and the Philippines [8,9]. The syndrome is diagnosed according to the 2003 Rotterdam Criteria if two of the following three criteria are met and other causes are excluded: (1) oligo- or anovulation, (2) clinical or laboratory signs of hyperandrogenemia (such as hirsutism, acne and alopecia), and (3) polycystic ovaries [10]. PCOS is associated with several comorbidities, including NAFLD but also insulin resistance, type 2 diabetes mellitus, metabolic syndrome, dyslipidemia, cardiovascular diseases, visceral obesity and certain types of endometrial cancer [11]. Despite its detrimental effect on long-term health, little attention has so far been paid to NAFLD as a comorbidity in PCOS. 

The pathogenesis of NAFLD in PCOS is complex. Major players that induce NAFLD in PCOS are assumed to be androgen access, chronic inflammation and insulin resistance [12]. In particular, the abundance of free fatty acids, the disbalance of adipokines, changes in the intestinal microbiota and inflammatory cytokines are common factors in the pathogenesis of insulin resistance and NAFLD and may thus play an important role in the development of NAFLD in PCOS [13,14]. For the diagnosis of NAFLD, a liver biopsy remains the gold standard. However, like any invasive diagnosis, it carries the risk of complications. Thus, less invasive methods, such as ultrasound or magnetic resonance imaging (MRI), are more commonly used to diagnose NAFLD. MRI has good sensitivity for detecting NAFLD and even quantifying it, but the high cost and low availability are limiting factors for its practical use [15]. Abdominal ultrasound, on the other hand, is generally less sensitive in obese patients. In addition, ultrasound can only reliably detect NAFLD when steatosis affects more than 20% of the total hepatocyte area. Therefore, the sensitivity in diagnosing milder forms of NAFLD by ultrasound is low [16]. 

In the past few years, surrogate markers have become more popular in the screening for NAFLD. One of these is the hepatic steatosis index (HSI), which is an easily available, yet reliable tool to screen for NAFLD. The HSI was developed from a cross-sectional study of approximately 5000 subjects with NAFLD and age- and sex-matched controls. In the analyses, the serum alanine aminotransferase (ALT) to serum aspartate aminotransferase (AST) ratio, a high body mass index (BMI) and diabetes mellitus were found to be easily accessible independent risk factors for NAFLD. The HSI was obtained by fitting these variables into a multiple regression model. Using the HSI, the sensitivity of ruling out NAFLD is 93% and the specificity of detecting NAFLD is 92% [17]. Compared with other scoring systems, the HSI yields a higher sensitivity and specificity and was validated for young adults, making it a suitable marker for assessing NALD in PCOS patients [18]. 

The role of metformin in the treatment of type 2 diabetes has been established for several decades. In addition, several other beneficial effects of metformin were described in the context of metabolic syndrome, cardiovascular disease, cancer or cognitive decline [19]. Metformin has been used as a pharmacological treatment in PCOS for many years. Multiple studies investigated its effect on several clinical outcomes. To summarize, metformin reduces weight and BMI in overweight patients [20]. It lowers the total testosterone but does not significantly improve hirsutism [6,21]. Interestingly, meta-analyses do not show improved fasting insulin or glucose under metformin treatment, whereas triglycerides did improve under metformin treatment. Due to the low quality of evidence across all outcomes, current guidelines give a “conditional recommendation” for PCOS treatment with metformin. The guidelines state that metformin treatment may be of greater benefit in higher metabolic risk groups. However, the current international guideline on PCOS does not address the high prevalence of NAFLD in PCOS patients [22]. 

As with the treatment of PCOS, there is no conclusive clinical evidence supporting the use of metformin in the treatment of NAFLD. Several studies found no beneficial effect of metformin on NAFLD or NASH (non-alcoholic steatohepatitis) in people with or without diabetes mellitus [23]. Other studies suggested a possible influence of metformin in improving NAFLD [24,25]. Nevertheless, clinical evidence is lacking to recommend metformin or any other pharmacological treatment for the management of NAFLD. While male gender and advanced age are major risk factors for the development of NAFLD [26], PCOS patients represent a diametrically different collective. We conducted this study in a first attempt to shed light on a possible effect of metformin on NAFLD in PCOS patients. We investigated whether metformin treatment could improve the hepatic steatosis index (HSI), which is a biomarker panel for assessing NAFLD, in the particular metabolic situation of PCOS. In order to address this question, we retrospectively analyzed HSI in PCOS patients at baseline without metformin treatment and at a follow-up visit after five to twelve months of metformin treatment. A control group consisted of PCOS patients who did not receive metformin treatment.

## 2. Materials and Methods

Patient data were retrospectively collected at a local outpatient clinic, namely, the endokrinologikum Göttingen, Von-Siebold-Strasse 3, 37075 Göttingen. In total, 403 files of patients who were diagnosed or suspected of having PCOS were evaluated for inclusion in either the metformin or the control group. For inclusion, the patients had to meet the following criteria: patients must (I) be diagnosed with PCOS, (II) be of reproductive age, and (III) have had a blood test including an oral glucose tolerance test and clinical examination at baseline t(0) and a blood test as well as a clinical examination at one follow-up visit five to twelve months after the baseline visit t(1). A diagnosis of PCOS was defined according to the Rotterdam criteria if two of the following three criteria were met and other causes were excluded: (1) oligo- or anovulation, (2) clinical or laboratory signs of hyperandrogenemia (such as hirsutism, acne and alopecia) and (3) polycystic ovaries [10]. Reproductive age was defined as between 14 to 45 years of age. The upper age limit was set to 45 instead of 49 to exclude perimenopausal situations as a possible confounder. Patients were included in the metformin group if they started their metformin treatment after the baseline visit and in the control group if they remained without treatment during the observation period. Each patient or their legal guardian filled out an informed consent form. We excluded all patients (I) whose diagnosis of PCOS was unsure or had to be withdrawn and (II) who had a known underlying liver disease, such as hepatitis of any type or who have been taking possibly hepatotoxic medication. None of the patients had diabetes according to their medical history and according to the glycemic values found during the oral glucose tolerance test. Chronic disease unrelated to hepatic pathologies, such as autoimmune thyroiditis, which is common in PCOS patients, was not considered as an exclusion criterion. Patients taking dietary supplements that are not considered hepatotoxic were also allowed in the study. 

At the time of the PCOS diagnosis, patients in our clinic received general advice on a healthy lifestyle, including a balanced diet and regular physical activity. Overweight and obese women were encouraged to follow a moderately calorie-reduced diet, with the goal of losing approximately 5–10% of their body weight. However, no specific dietary and physical activity measures were documented in the patient records; therefore, they could not be included in the study. After re-evaluation of the patients according to the stated criteria, 81 patients could be included in the study, of which 68 had begun a treatment with metformin after the baseline visit and 13 had not. The usual dosage for metformin was 850 mg twice daily; however, some patients with lower body weight received 500 mg twice daily and some with higher body weight received 1000 mg twice daily. The patients within the metformin group were further divided into three groups according to their initial body mass index (BMI): a non-obese group with an initial BMI < 30 kg/m^2^ (*n* = 24), an obese group with an initial BMI 30–35 kg/m^2^ (*n* = 17) and a very obese group with an initial BMI >35 kg/m^2^ (*n* = 27).

The blood test at baseline and at the follow-up visit included the following parameters: AST, ALT, fasting glucose and fasting insulin. In addition, baseline data included triglycerides, high-density lipoprotein (HDL) and an oral glucose tolerance test (OGTT) with an insulin measurement. In order to assess hepatic insulin resistance, we used the homeostatic model assessment for insulin resistance (HOMA-IR). The HOMA-IR was calculated at baseline and the follow-up visit according to the following formula: $$\mathrm{HOMA-IR}\;=\frac{\mathrm{fasting}\;\mathrm{insulin}\;\left(\mathrm{mU}/L\right)\times \mathrm{fasting}\;\mathrm{glucose}\;\left(\mathrm{mg}/\mathrm{dL}\right)}{405}$$

Values > 2.5 were regarded as suggestive of insulin resistance [27]. Whole-body insulin sensibility was assessed using the Matsuda index. The Matsuda index was calculated at the baseline visit based on the results of the OGTT using the following formula [28]: $$\mathrm{Matsuda}\;\mathrm{index}\;=\frac{10000}{\sqrt{\left(\mathrm{fasting}\;\mathrm{glucose}\times \mathrm{fasting}\;\mathrm{insulin}\right)\times \left(\mathrm{mean}\;\mathrm{glucose}\times \mathrm{mean}\;\mathrm{insulin}\right)}}$$

Mean glucose during the OGTT was calculated as
$$\mathrm{mean}\;\mathrm{glucose}\;=\frac{\mathrm{gluc}0\;\min+2\times \mathrm{gluc}60\;\min+\;\mathrm{gluc}\;120\;\min}{4}$$

Mean insulin during the OGTT was calculated as
$$\mathrm{mean}\;\mathrm{insulin}\;=\frac{\mathrm{ins}0\;\min+2\times \mathrm{ins}60\;\min+\;\mathrm{ins}\;120\;\min}{4}$$

A Matsuda index of <2.5 was regarded as suggestive of insulin resistance. Clinical parameters were available from the patients’ records at baseline t(0) and at the follow-up visit t(1). They included height, weight, blood pressure (only at baseline) and age. 

We calculated the hepatic steatosis index (HSI) retrospectively for each patient at the baseline visit t(0) and the follow-up visit t(1). The HSI was calculated using the following formula: $$\mathrm{HSI}\;=8\times \frac{\mathrm{ALT}}{\mathrm{AST}}+\mathrm{BMI}+2\left(\mathrm{if}\;\mathrm{type}\;2\;\mathrm{diabetes}\;\mathrm{is}\;\mathrm{yes}\right)+2\;\left(\mathrm{if}\;\mathrm{patient}\;\mathrm{is}\;\mathrm{female}\right)$$

The HSI was defined with values <30 basically ruling out NAFLD and values >36 being highly suggestive of NAFLD [17].

### Statistical Analyses

Differences between characteristics in the treatment group vs. the control group were tested using the Mann–Whitney U for non-parametric tests. Fisher’s exact test was performed for the analysis of contingency tables with a small sample size. Differences between the BMI or HSI at t(1) vs. t(0) were tested using Wilcoxon tests for paired non-parametric analyses. Differences in baseline characteristics between the BMI groups were analyzed using Kruskal–Wallis tests for non-parametric group analyses. Any *p*-values < 0.05 were considered statistically significant. All analyses were carried out using IBM^®^ SPSS^®^ Statistics 27 (IBM Corp., Armonk, NY, USA).

## 3. Results

### 3.1. Baseline Characteristics of PCOS Patients Receiving Metformin Treatment Did Not Differ from Control Patients

The data of 81 outpatient women with PCOS were retrospectively analyzed in order to elucidate whether metformin treatment affects the hepatic steatosis index (HSI). Sixty-eight of the women received metformin treatment after their baseline visit (metformin group) and 13 did not (control group). We compared the baseline characteristics of the metformin and control groups in order to see whether there was a bias as to who received the metformin treatment (Table 1). Patients in both groups were similar in age, body weight and BMI. Lab parameters, including triglycerides, HDL cholesterol, fasting glucose and insulin levels, and liver enzymes, were also similar in both groups, except for AST levels, which were lower in the control group (21.6 ± 1.9 vs. 32.1 ± 3.1 U/L, *p* < 0.01). Hepatic insulin resistance was estimated using the HOMA-IR and whole-body insulin sensitivity was estimated using the Matsuda index. Neither index differed between the groups at baseline. The mean levels of the liver enzymes ALT and AST were within the normal ranges in both groups. However, AST was significantly lower in the control group. The mean HSI was clearly pathologic in both groups and reached 43.2 ± 1.0 in the metformin group and 41.9 ± 3.3 in the control group. 

### 3.2. HSI Significantly Improved after Seven Months of Metformin Treatment

The hepatic steatosis index (HSI) improved significantly from 43.2 ± 1.0 to 41.0 ± 1.1 (*p* < 0.0001) in women who received the metformin treatment (Figure 1a). The absolute number of women with a pathological HSI of >36 decreased with metformin treatment from 53/68 (77.9%) to 44/68 (65.0%) but slightly increased in the control group from 7/13 (53.8%) to 8/13 (62.0%). Given the HSI formula (HSI = 8 × (ALT/AST) + BMI + 2 (if type 2 diabetes is yes) + 2 (if the patient is female)), changes in liver enzymes may affect the HSI as much as changes in BMI. Therefore, we further analyzed the impact of metformin treatment on the liver enzymes, ALT and AST, as well as on BMI individually. In line with the improved HSI, the metformin group lowered their BMI from 33.0 ± 0.9 to 31.2 ± 0.8 kg/m^2^ (*p* < 0.0001) and their body weight from 90.7 ± 2.5 to 86 ± 2.4 kg (*p* < 0.0001) (Figure 1c,d). Liver enzymes also improved significantly after the metformin treatment, with ALT levels decreasing from 32.1 ± 3.1 to 27.6 ± 2.2 U/L (*p* < 0.05) and AST levels decreasing from 29.6 ± 1.4 to 26.8 ± 1.2 U/L (*p* < 0.01) (Figure 1e,f). In addition, a possible influence of metformin treatment on hepatic insulin resistance was evaluated by assessing the HOMA-IR. Women in both the control and the metformin groups had a pathological HOMA-IR at baseline as an indicator for hepatic insulin resistance. Despite the metformin treatment, the HOMA-IR significantly increased from 4.1 ± 0.4 to 4.6 ± 1.2 (*p* < 0.05) (Figure 1b). The HSI, HOMA-IR, BMI, body weight and liver enzymes remained unchanged in the control group during the observation period (Figure 1a–f).

### 3.3. Baseline Parameters in BMI Subgroups

Current guidelines for the treatment of PCOS recommend metformin mainly in patients with increased metabolic risk [22]. Therefore, we investigated whether different BMI groups showed different HSI responses to metformin treatment. The 68 patients with PCOS and metformin treatment were divided into three BMI groups in order to analyze their responses to metformin. The baseline characteristics of the patients are shown in Table 2. The mean BMI was 25.6 ± 0.6 kg/m^2^ in the non-obese group (BMI < 30 kg/m^2^), 32.2 ± 0.4 kg/m^2^ in the obese group (BMI 30–35 kg/m^2^) and 40.2 ± 0.7 kg/m^2^ in the very obese group (BMI > 35 kg/m^2^). The age of the patients was similar in all groups. Fasting glucose did not differ between the groups. However, the more obese patients had higher insulin levels, higher triglyceride levels and tended to have lower HDL levels than the less obese patients. Furthermore, diastolic blood pressure differed significantly between the groups with 85 ± 1 mmHg in the non-obese, 89 ± 3 mmHg in the obese and 94 ± 3 mmHg in the very obese group (*p* < 0.01). The less obese patients received a lower absolute daily dosage of metformin (1507 ± 73 vs. 1622 ± 71 vs. 1770 ± 28 mg); however, the dosage per kg body weight was higher (21.5 ± 0.8 vs. 18.7 ± 0.7 vs. 16.2 ± 0.4 mg/kg body weight). The intake of oral contraceptives was equally low in all weight groups (Table 2). 

### 3.4. BMI Significantly Improved after Metformin Therapy in All Weight Groups

At the time of PCOS diagnosis, patients were routinely informed that weight loss has a positive effect on PCOS. In addition, metformin treatment can support weight loss in PCOS patients [20]. In our study, PCOS patients significantly improved their BMI after metformin treatment throughout all weight groups. The absolute reduction in BMI was especially pronounced in very obese women from 40.2 ± 3.5 kg/m^2^ to 37.6 ± 5.1 kg/m^2^ (*p* < 0.001). In obese women, the BMI improved from 32.2 ± 1.5 kg/m^2^ to 30.5 ± 2.3 kg/m^2^ (*p* = 0.002), and in non-obese women from 25.6 ± 3.1 kg/m^2^ to 24.4 ± 2.9 kg/m^2^ (*p* < 0.001) (Figure 2). 

### 3.5. Metformin Treatment Significantly Improved HSI in Normal or Overweight but Not in Obese and Very Obese Patients with PCOS

As described above, in most PCOS patients, the HSI was elevated to a level that suggested the presence of NAFLD to be very likely (HSI > 36) (Table 1 and Table 2). HSI was most elevated in the very obese group (51.2 ± 4.7). Obese women also showed a clearly pathologic HSI of 41.2 ± 2.7, whereas the steatosis index was borderline in non-obese women (35.5 ± 4.5). While the baseline HSI clearly followed the BMI pattern, data after the metformin treatment showed a different picture. Although the effect of metformin treatment on the BMI was smallest in the non-obese group, these patients showed a significant reduction in HSI following the metformin treatment by 1.5 ± 2.1 (*p* < 0.001). Despite the greater weight loss in the obese and very obese groups, the HSI did not significantly change in these groups after the metformin therapy (Figure 3). 

## 4. Discussion

Non-alcoholic fatty liver disease (NAFLD) is highly prevalent in PCOS patients. In our cohort, 60/81 (74.1%) of the patients had a hepatic steatosis index (HSI) > 36, which is considered very suggestive of NAFLD. Using transient elastography in a cross-sectional cohort study, Shengir et al. found NAFLD or hepatic fibrosis in 46.5% of women with PCOS [29]. A similar study even found NAFLD in 84.3% of the patients if hyperandrogenism, ovulatory dysfunction and polycystic ovaries were present [2]. Thus, our finding of high NAFLD prevalence according to the HSI is similar to the prevalence found in the abovementioned elastography studies on PCOS patients. This underscores the possible value of HSI as a readily accessible surrogate parameter for non-alcoholic fatty liver disease in PCOS patients. 

The pathomechanism involved in the development of NAFLD in PCOS patients is still a subject of research. Insulin resistance, adipose tissue dysfunction, hyperandrogenemia and inflammation are factors that are probably involved [30]. A deeper understanding of the pathophysiological mechanisms leading to NAFLD in PCOS could point the way to pharmacological treatment options for this disease. In other metabolic pathologies that are associated with NAFLD, such as obesity and type 2 diabetes, favorable effects of metformin on NAFLD were described. However, the clinical evidence for the beneficial effects of metformin is not strong enough to define metformin as a treatment option for NAFLD. Therefore, to date, the only proven effective therapies for NAFLD remain physical exercise and weight reduction [31]. However, metformin was shown to have direct effects on hepatic lipid metabolism at the cellular level. Metformin-induced activation of the AMP kinase-dependent pathway inhibits de novo lipogenesis on the one hand and increases fatty acid oxidation on the other hand. Both effects may alleviate liver steatosis [32,33]. In addition, metformin was shown in animal studies to increase liver leptin sensitivity in a dose-dependent manner [34]. Despite these effects at a cellular level, solid evidence for metformin as an effective treatment for NAFLD is still lacking in the clinical setting. 

Given the extremely high prevalence of NAFLD in PCOS and the long-term detrimental effect of NAFLD on hepatic and cardiovascular health in these women [5,14], it is essential to pave the way for future treatment options. Because of the positive effects of metformin on hepatic lipid metabolism [32] but also on insulin resistance [35], adipose tissue remodeling [35,36] and inflammation [37], we hypothesized that NAFLD might improve with metformin treatment in women with PCOS. Using the HSI as a surrogate parameter for NAFLD, we found a significant improvement in the HSI in women with PCOS after approximately seven months of metformin treatment. Owing to the retrospective character of the study, we cannot make a conclusive statement about the direct effect of metformin treatment on NAFLD in PCOS. Since we observed a significant reduction in BMI following the metformin treatment, we sought to understand whether weight loss alone was responsible for the observed improvement in the HSI. In particular, the liver enzyme ALT is considered a good marker for hepatic damage [38] and both AST and ALT were shown to be significantly associated with steatohepatitis [39]. In our study, both AST and ALT were found to decrease with metformin treatment. This is in line with an earlier study that found improved GGT and ALT levels in PCOS women after a similar time of metformin treatment [40]. As mentioned before, besides exercise, weight loss is the only proven therapy for NAFLD [31]. Therefore, the weight reduction observed in our patients can certainly be interpreted as an important aspect that may improve NAFLD. In addition, the observed reduction in ALT but also AST in metformin-treated women may reflect a direct effect of metformin on the hepatocellular level, which was described in [32,33,34]. Thus, metformin may exert its effects on NAFLD in PCOS, both through a reduction in body weight but also through direct hepatic effects. 

In order to find out whether certain subgroups of PCOS patients may particularly benefit from metformin treatment in terms of liver health, we analyzed the HSI in the BMI subgroups before and after the metformin treatment. Interestingly, the HSI improved in non-obese women with PCOS (BMI < 30 kg/m^2^), while obese and very obese women with BMIs of 30–35 or >35 kg/m^2^, respectively, did not show significant improvements. This finding was notable because the obese and very obese women treated with metformin in our cohort had particularly pronounced weight loss, and, as stated above, weight loss is an important factor that can improve NAFLD. From our data, we learned that non-obese women with PCOS may particularly benefit from metformin treatment with regard to fatty liver disease. One reason for the more pronounced effect of metformin in leaner patients might be the relatively higher dosing of metformin in this weight group. While absolute dosing was higher in the more obese patients, the less obese women received higher doses of metformin per kg body weight. As described, metformin has effects on hepatic lipid metabolism and leptin receptor signaling, which can be dose-dependent [32,34]. In obese and very obese women with PCOS, very high hepatic steatosis indices may reflect more advanced liver disease. Possibly, these advanced stages of NAFLD require a more effective treatment, which could be achieved using a longer treatment duration or a different pharmacologic agent. 

This observational study had limitations. First, the number of patients per group was relatively small such that slight effects may not be detected. Second, the observational period of about seven months of metformin therapy may not be sufficiently long to notably improve more advanced stages of fatty liver disease. Third, our findings were limited to the retrospective evaluation of non-imaging biomarkers as an indirect assessment of NAFLD. In future prospective studies, the hepatic health of PCOS patients should be addressed not only by laboratory and clinical parameters but also by imaging tools, such as abdominal ultrasound, fibroscan or elastography of the liver. These methods allow for non-invasively assessing the degrees of hepatic steatosis or fibrosis. However, imaging tools also have limitations, particularly in obese patients. Therefore, laboratory tools to evaluate hepatic steatosis or fibrosis will continue to be important, especially in overweight patients. It will also be interesting to see how the HSI develops in women with PCOS who receive a different type of medical therapy. In this context, it is of particular interest to investigate the effect of a GLP-1 analog on NAFLD in PCOS. One prospective trial showed that 26 weeks of liraglutide treatment was associated with a reduction in liver fat measured using HMR spectroscopy in women with PCOS. Similar to our retrospective findings after metformin treatment, significant weight loss was also observed under liraglutide treatment. However, the study did not discuss whether the perceived effect on liver fat was a secondary effect of the weight loss or could have been a direct effect of the liraglutide treatment [41]. In a phase 2 trial, Newsome et al. demonstrated an effect of semaglutide on nonalcoholic steatohepatitis [42]. The effect of semaglutide on NAFLD has not been studied in women with PCOS and could be an interesting subject of future research.

Despite the limitations due to the relatively small sample size and the biomarker-based approach in our study, the extremely high prevalence of pathological HSIs strongly suggested that non-alcoholic fatty liver disease (NAFLD) was a significant problem in patients with PCOS. It is known that NAFLD may progress to liver fibrosis, liver cirrhosis and even hepatocellular carcinoma. NAFLD is also associated with elevated cardiovascular disease. We feel that it is essential to address the liver situation in patients with PCOS and to find ways to improve NAFLD in these patients. Our data suggested that metformin may have had a beneficial effect on NAFLD in normal and overweight women with PCOS. Prospective studies are needed to further investigate the possible effect of metformin on NAFLD in these young female patients. Future studies should use a combination of laboratory and imaging tools and should address the question of whether longer periods of metformin or the use of other drugs may have an effect on obese women with PCOS. 

## Figures and Tables

**Figure 1 jcm-11-04294-f001:**
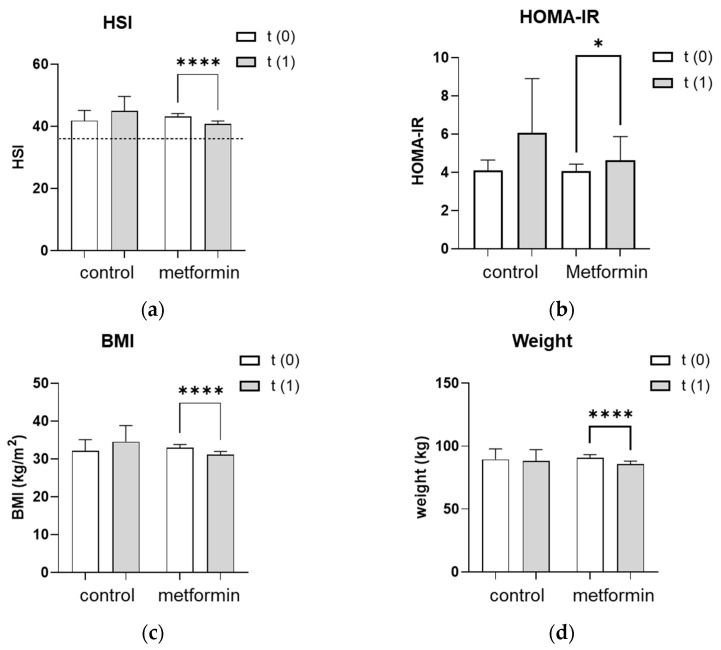
Effect of seven months of metformin treatment. (**a**–**f**) Metabolic and liver parameters were analyzed at baseline (t(0)) and after an average of seven months without (control) or with (metformin) metformin treatment (t(1)). (**a**) The dotted line represents the cutoff of 36 for a pathologic HSI. (**a**–**f**) Differences between characteristics in the metformin group vs. the control group were tested using the Mann–Whitney U for non-parametric tests. Differences between parameters at t(1) vs. t(0) were tested using Wilcoxon tests for paired non-parametric analyses. Data are presented as mean ± standard error of the mean. * *p* < 0.05, ** and †† *p* < 0.01, **** *p* < 0.0001. HSI: hepatic steatosis index, HOMA-IR: homeostatic model assessment for insulin resistance, BMI: body mass index, ALT: alanine transaminase, AST: aspartate aminotransferase.

**Figure 2 jcm-11-04294-f002:**
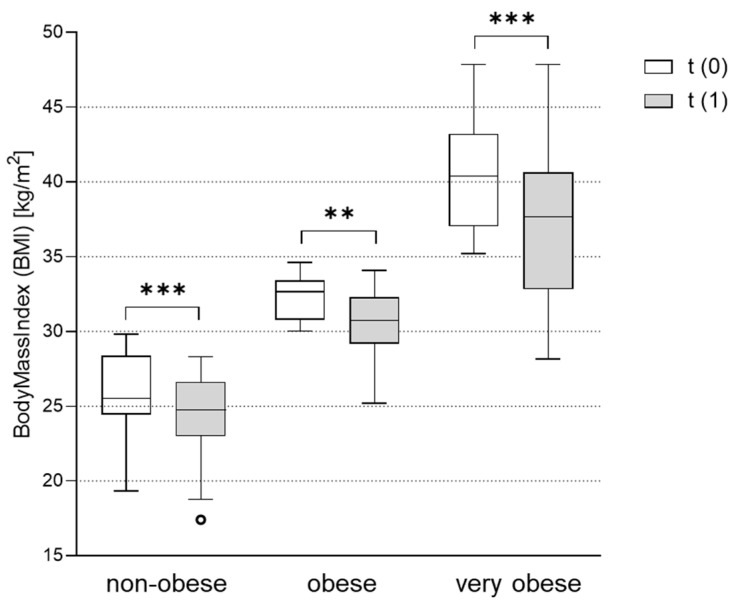
Effect of metformin treatment on body mass index (BMI) in the non-obese, obese and very obese patients. BMI was assessed in subgroups of non-obese (BMI < 30 kg/m^2^), obese (BMI 30-35 kg/m^2^) and very obese (BMI > 35 kg/m^2^) women with PCOS before (t(0)) and after (t(1)) treatment with metformin. Differences between BMI at t(1) vs. t(0) were tested using Wilcoxon tests for paired non-parametric analyses. Data are presented as mean ± standard error of the mean. ** *p* < 0.01, *** *p* < 0.001. BMI: body mass index.

**Figure 3 jcm-11-04294-f003:**
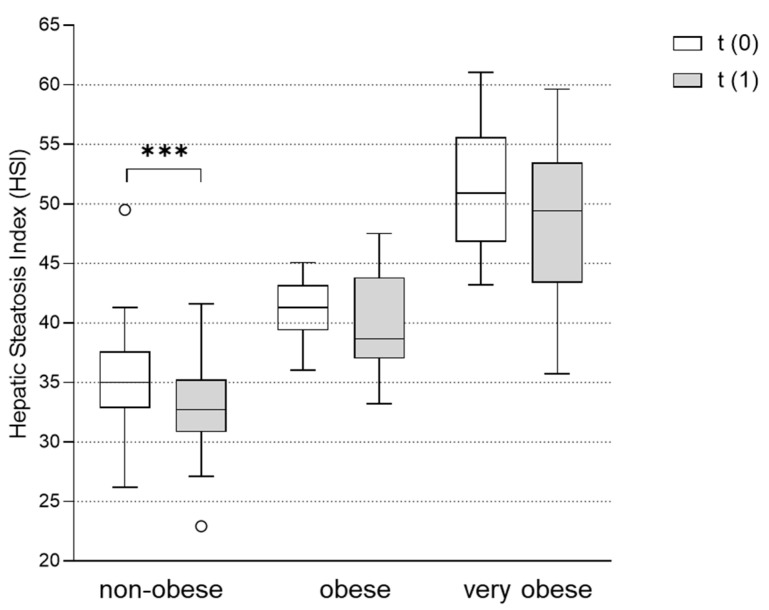
Effect of metformin treatment on hepatic steatosis index (HSI) in non-obese, obese and very obese patients. The HSI was assessed in subgroups of non-obese (BMI < 30 kg/m^2^), obese (BMI 30-35 kg/m^2^) and very obese (BMI > 35 kg/m^2^) women with PCOS before (t(0)) and after (t(1)) treatment with metformin. Differences between the HSI at t(1) vs. t(0) were tested using Wilcoxon tests for paired non-parametric analyses. Data are presented as mean ± standard error of the mean. *** *p* < 0.001. HSI: hepatic steatosis index.

**Table 1 jcm-11-04294-t001:** Baseline characteristics of the control and metformin groups.

	Control Group Mean ± SEM or *n* (%)	Metformin Group Mean ± SEM or *n* (%)	*p*-Value
Number	13	68	
Clinical parameters			
Age (years)	25.3 ± 1.4	26.3 ± 0.8	0.81
Body weight (kg)	89.4 ± 8.4	90.7 ± 2.5	0.65
Body mass index (kg/m^2^)	32.1 ± 3.0	33.0 ± 0.9	0.64
Systolic blood pressure (mmHg)	137.5 ± 4.6	137.7 ± 2.3	0.19
Diastolic blood pressure (mmHg)	85.2 ± 2.1	89.3 ± 1.4	0.98
Co-medication			
Oral contraceptive	1/13 (8%)	9/68 (13%)	1
Laboratory parameters			
Triglycerides (mg/dL)	133.5 ± 29.5	115.4 ± 6.9	0.93
HDL-cholesterol (mg/dL)	58.3 ± 7.7	55.9 ± 2.3	0.88
Fasting glucose (mg/dL)	87.9 ± 2.9	84.4 ± 2.0	0.79
Fasting insulin (µU/mL)	20.2 ± 2.1	14.2 ± 2.2	0.79
HOMA-IR	4.1 ± 0.6	4.1 ± 0.4	0.59
Matsuda index	2.9 ± 0.7	3.3 ± 0.3	0.46
ALT (U/L)	21.6 ± 1.9	32.1 ± 3.1	0.08
AST (U/L)	22.0 ± 1.2	29.6 ± 1.4	<0.01 *
HSI	41.9 ± 3.3	43.2 ± 1.0	0.51
HSI > 36	7/13 (53.8%)	53/68 (77.9%)	0.07

Mann–Whitney U test was used for comparing baseline data in the metformin vs. control groups, and chi-square and Fisher’s exact tests were used to compare the number of patients. SEM: standard error of the mean, HDL: high-density lipoprotein, HOMA-IR: homeostatic model assessment for insulin resistance, ALT: alanine transaminase, AST: aspartate aminotransferase, HSI: hepatic steatosis index, * difference between the groups reached statistical significance.

**Table 2 jcm-11-04294-t002:** Baseline characteristics of the patients in different BMI groups.

	Group I: BMI < 30 kg/m^2^	Group II: BMI 30–35 kg/m^2^	Group III: BMI > 35 kg/m^2^	*p*-Value *
Number	24	17	27	
Observational time (months)	6.4 ± 0.3	7.3 ± 0.3	7.3 ± 0.3	
Demographic variables				
Age (years)	25.5 ± 1.5	26.4 ± 1.7	27.0 ± 1.1	0.73
Body weight (kg)	70.8 ± 2.3	87.1 ± 1.5	110.6 ± 2.2	<0.001 *
BMI (kg/m^2^)	25.6 ± 0.6	32.2 ± 0.4	40.2 ± 0.7	<0.001 *
Systolic blood pressure (mmHg)	132 ± 3	136 ± 5	143 ± 4	0.26
Diastolic blood pressure (mmHg)	85 ± 1	89 ± 3	94 ± 3	<0.01 *
Medication				
Metformin dosage (mg/day)	1507 ± 73	1622 ± 71	1770 ± 28	<0.01 *
Metformin dosage (mg/kg body weight)	21.5 ± 0.8	18.7 ± 0.7	16.2 ± 0.4	<0.001 *
Oral contraceptive	3/24	2/17	4/27	1
Laboratory parameters				
Triglycerides (mg/dL)	90.8 ± 8.3	119.7 ± 10.5	134.8 ± 13.0	0.02 *
HDL-cholesterol (mg/dL)	62.4 ± 3.5	53.6 ± 4.4	51.7 ± 4.0	0.12
Fasting glucose (mg/dL)	84.4 ± 2.0	91.1 ± 3.7	87.9 ± 2.9	0.31
Fasting insulin (µU/mL)	14.2 ± 2.2	20.4 ± 2.1	20.2 ± 2.1	0.10
HOMA index	2.9 ± 0.5	4.8 ± 0.7	4.4 ± 0.5	0.04 *
Matsuda index	3.9 ± 0.7	2.5 ± 0.5	2.5 ± 0.2	0.06
ALT (U/L)	30,2 ± 6.7	26.2 ± 3.1	37.6 ± 4.6	0.06
AST (U/L)	27.3 ± 2.5	28.4 ± 1.8	32.4 ± 2.4	0.17
HSI	35.5 ± 0.9	41.2 ± 0.6	51.2 ± 0.9	<0.001 *
HSI > 36	9/24	17/17 (100%)	27/27 (100%)	<0.001 *

Data are represented as mean value ± standard error of the mean. Kruskal–Wallis group comparison for non-parametric distribution was used to identify differences between the groups. HDL: high-density lipoprotein, HOMA-IR: homeostatic model assessment for insulin resistance, ALT: alanine transaminase, AST: aspartate aminotransferase, HSI: hepatic steatosis index, * difference between the groups reached statistical significance.

## Data Availability

Data is contained within the article.

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
