# Peer review of "Metformin Improves the Hepatic Steatosis Index in Non-Obese Patients with Polycystic Ovary Syndrome"

_jcm, 2022, doi:10.3390/jcm11154294_

Round 1
Reviewer 1 Report
No other comments
Author Response
We thank the reviewer for taking the time to review the manuscript and are pleased that he/she has approved it in its present form.
Reviewer 2 Report
The manuscript is interesting. However, this reviewer raises some issues that need to be addressed by authors.
1- Methods. It is not clear whether diabetes is an exclusion criterion for patient recruitment. On the other hand, neither the glycemic values at 120 minutes after the oral load test with 75 grams of glucose nor the HbA1c are indicated in the text. Therefore, I wonder how diabetes can be ruled out in the population studied.
2- Was a targeted family history for diabetes mellitus performed?
3- In the text, formulas should be formatted in a way that is clearer to the reader.
4- NAFLD and PCOS share the central pathophysiological role played by insulin resistance. Two very recent reviews explain in an updated and complete way these pathophysiological mechanisms in NAFLD (Antioxidants, 2021, 10(2), pp. 1–25, 270. doi: 10.3390/antiox10020270. - Processes, 2021, 9(1), pp. 1–18, 135. doi: 10.3390/pr9010135), Authors should address this issue and add the above references to the manuscript.
5- Finally, in recent years metformin has shown several extra-glycemic positive health effects. In particular, evidence has been accumulated on a beneficial impact against many aging-related morbidities as obesity, metabolic syndrome, cardiovascular disease, cancer, cognitive decline and mortality. A recent review focused all these new effects of metformin (Diabetes Res Clin Pract. 2020 Feb;160:108025. doi: 10.1016/j.diabres.2020.108025.). Authors should add and comment on this reference, and also refer, albeit briefly, to the other extra-antidiabetic effects of metformin.
Round 2
Reviewer 2 Report
No further comments.
This manuscript is a resubmission of an earlier submission. The following is a list of the peer review reports and author responses from that submission.
Round 1
Reviewer 1 Report
I read this study by Riemann et al. with great interest. The study aimed to evaluate the Metformin may improve Non-Alcoholic Fatty Liver Disease in non-obese Patients with Polycystic Ovary Syndrome: a retrospective Analysis of the Hepatic Steatosis Index in PCOS. The result of this study postulated that a significant improvement in HSI was observed after metformin treatment in non-obese women. Despite of its great contribution to expand our current knowledge, this idea suffers from several weaknesses to be used to make a valid and trustworthy conclusion.
- Define acronyms and abbreviations at first use.
- please discuss about your novelty.
- It is required to provide more information about Materials and Methods. However, the sample size is low and the conclusion should be interpreted with caution.
- Please mention Rotterdam criteria in the Materials and Methods.
- Please give your reasons why you did not evaluate the chronic disease and taking dietary supplements as exclusion criteria. Please explain about inclusion and exclusion criteria.
- Given the variation of metformin doses among groups, could it consider a confounding factor?
- It has been suggested to evaluate HbA1C as a more reliable marker of long-term variations in blood glucose.
- Please provide reference for this sentence “Thus, our finding of high NAFLD prevalence according to HSI is consistent with elastography studies on PCOS patients.”
- Did participant give any advice about diet therapy and physical activity? Please clarify the reason of not evaluating the physical activity in this study. Because diet therapy and physical activity are considered to be confounding factors.
- Please describe the possible reasons for choosing 14-18 years of age as your inclusion criteria.
- According to the results, the discussion is not well-described and there are insufficient shreds of evidence to support it. Besides, there needs to be more evidence to prove the results. It is a major issue in this paper.
- The overall novelty of the study is dubious in terms of making a significant advance in our knowledge. It’s a big problem.
Author Response
We would like to thank the reviewer for the very helpful comments, which we have addressed in the attached file.

Reviewer 2 Report
1 nimer of patients is too small
2 mechanism of fat. Improvement by method in
3 fibroscan or ARFI before and after metformin
5 duration of treatment need to be longer
6 please consider the addition of demagnetise for
Author Response

(The authors gave the same response as above.)

Round 2
Reviewer 1 Report
Dear Sir
Despite its great contribution to expand our current knowledge, this idea suffers from several weaknesses to be used to make a valid and trustworthy conclusion